# Luminescent Aptamer-Based Bioassays for Sensitive Detection of Food Allergens

**DOI:** 10.3390/bios12080644

**Published:** 2022-08-15

**Authors:** Donato Calabria, Martina Zangheri, Seyedeh Rojin Shariati Pour, Ilaria Trozzi, Andrea Pace, Elisa Lazzarini, Maria Maddalena Calabretta, Mara Mirasoli, Massimo Guardigli

**Affiliations:** 1Department of Chemistry “Giacomo Ciamician”, Alma Mater Studiorum, University of Bologna, Via Francesco Selmi 2, I-40126 Bologna, Italy; 2Interdepartmental Centre for Industrial Aerospace Research (CIRI AEROSPACE), Alma Mater Studiorum, University of Bologna, Via Baldassarre Canaccini 12, I-47121 Forlì, Italy; 3Interdepartmental Centre for Industrial Agrofood Research (CIRI AGRO), Alma Mater Studiorum, University of Bologna, Via Quinto Bucci 336, I-47521 Cesena, Italy; 4Interdepartmental Centre for Industrial Research in Advanced Mechanical Engineering Applications and Materials Technology (CIRI MAM), Alma Mater Studiorum, University of Bologna, Viale Risorgimento 2, I-40136 Bologna, Italy; 5Interdepartmental Centre for Industrial Research in Renewable Resources, Environment, Sea, and Energy (CIRI FRAME), Alma Mater Studiorum, University of Bologna, Via Sant’Alberto 163, I-48123 Ravenna, Italy

**Keywords:** aptamer, allergen, biosensor, chemiluminescence, fluorescence, food

## Abstract

The presence of hidden allergens in food products, often due to unintended contamination along the food supply chain (production, transformation, processing, and transport), has raised the urgent need for rapid and reliable analytical methods for detecting trace levels of such species in food products. Indeed, food allergens represent a high-risk factor for allergic subjects due to potentially life-threatening adverse reactions. Portable biosensors based on immunoassays have already been developed as rapid, sensitive, selective, and low-cost analytical platforms that can replace analyses with traditional bench-top instrumentation. Recently, aptamers have attracted great interest as alternative biorecognition molecules for bioassays, since they can bind a variety of targets with high specificity and selectivity, and they enable the development of assays exploiting a variety of transduction and detection technologies. In particular, aptasensors based on luminescence detection have been proposed, taking advantage of the development of ultrasensitive tracers and enhancers. This review aims to summarize and discuss recent efforts in the field of food allergen analysis using aptamer-based bioassays with luminescence detection.

## 1. Introduction

Food allergies are a principal issue about food safety in industrialized countries. Indeed, the presence of undeclared allergens has been identified as the leading recorded cause of food safety incidents/recalls in recent years [1]. To avoid any accidental exposure to allergenic substances, EU Regulation No. 1169/2011 imposes a detailed declaration of allergens, even if present in traces, in the list of ingredients of prepacked foods, as well as complete allergen information for non-prepacked foods, including those served in restaurants and cafes.

The major trend in preventing adverse reactions is to implement manufacturing processes for removing allergenic compounds in food. However, the unintended contamination of foodstuff with allergen sources can occur during the manufacturing processes (i.e., production, transformation, packaging, and transportation) [2]. For this reason, recent research in food safety has been focused on developing rapid, sensitive, and user-friendly methods for the recognition and detection of hidden allergens in complex food matrices. Biosensors are a promising approach, since they combine the high selectivity of biorecognition elements, either natural (e.g., antibodies) or synthetic (e.g., aptamers or imprinted polymers), with simple preanalytical and analytical procedures [3,4,5,6,7]. Portable biosensors are ideal for food safety applications, as attested by several works reported in the literature for the detection of pathogens [8,9], toxins [10,11], pesticides and veterinary drug residues [12,13], organic pollutants [14,15], and allergens [16].

Portable biosensing devices are based on the integration of three components: the biospecific element that recognizes the target analyte, the transducer that converts the biorecognition event into a measurable chemical–physical signal, and the signal processing device that elaborates the signal and provides the result. For each component, there are several alternatives that can be selected, depending on the analytical application for which the biosensor is developed. Indeed, the biorecognition element could be an antibody raised against the target analyte, a single-strand DNA molecule suitable for hybridizing with a target DNA fragment, an aptamer that recognizes the target, an enzyme that catalyzes a specific reaction involving the target analyte, or even a living cell or tissue able to detect the analyte through physiological or biochemical processes. 

Detecting traces of allergens in foodstuffs is a hard analytical task, as it is necessary to identify small quantities of the target analyte in a complex and heterogeneous matrix containing various structurally similar components that can act as interferents. Considering these requirements, the bioreceptor performance is crucial. Antibodies have been used for a long time, and they are still the gold standard for the development of food safety biosensors, such as lateral flow immunoassays (LFIAs). The LFIA platform relies on the use of nitrocellulose membrane strips that are functionalized in specific areas with immunoreagents able to recognize and capture the target analyte. Since the sample and the immunoreagents flow through the strip by capillarity and the final reading can be often performed by the naked eye (e.g., using immunoreagents labeled with gold nanoparticles), this platform is attractive for its simplicity and low cost [17,18]. In addition to the many examples reported in the literature [19], many LFIAs for food allergens are commercially available for routine application (Table 1). All the LFIA-based devices reported are based on a visual colorimetric detection for a qualitative evaluation of the presence of the allergens of interest in raw materials or finished products.

Recently, a great deal of attention has been paid to aptamers, i.e., DNA or RNA oligonucleotide sequences designed to bind specifically to a given target molecule. Aptamers have found applications in analytical methods for environmental [27,28], food safety [29,30], and healthcare [31,32] purposes. Differently from antibodies, aptamers are synthetic molecules obtained in vitro, mostly by the SELEX process. SELEX (Systematic Evolution of Ligands by EXponential enrichment) is an iterative process for the selection of high-affinity aptamers within a large pool of randomly generated oligonucleotide sequences. The first iteration begins with incubating the target analyte with a library of randomized oligonucleotide sequences. The unbound aptamer sequences are washed away, and the bound aptamers are collected and amplified through polymerase chain reaction (PCR) to generate the oligonucleotide library for the next SELEX iteration. By increasing the stringency of the target recognition reaction, the tightest-binding sequences are identified through the subsequent rounds of selection [33]. Alternatively, non-SELEX methods can be employed, such as nonequilibrium capillary electrophoresis of equilibrium mixture (NECEEM)-based partitioning, which consists of a sequence of cycles, each including an incubation of the randomized oligonucleotide library with the target, followed by the separation of the bound from the unbound nucleic acid ligands without the need for amplification [34]. The approaches for the selection of target-specific aptamers based on the screening of randomized oligonucleotide libraries have been integrated by bioinformatics tools for the in silico prediction of aptamer–protein interactions, which can help to identify novel target-selective aptamers and deepen the knowledge of the interaction between the aptamer and target [35,36]. A selection of oligonucleotide aptamers for the detection of food allergens described in the literature are reported in Table 2.

While antibodies can be produced against immunogenic molecules, aptamers can be targeted at a vast range of targets, from small molecules to cells. Furthermore, when compared with antibodies, nucleic acid aptamers are stable in a wider range of temperatures, they have longer shelf lives and higher batch-to-batch reproducibility, and they can easily be reused after denaturation and produced at relatively low cost [48]. A comparison between the approaches for producing aptamers and antibodies is reported in Figure 1. Finally, aptamers are easily marked with different labels [49,50], allowing the employment of a variety of transduction systems. Among them, optical detection methods based on luminescent phenomena such as fluorescence (FL), chemiluminescence (CL), bioluminescence (BL), and electrochemiluminescence (ECL) have attracted great attention due to their high sensitivity combined with the possibility to easily implement dedicated detectors into portable integrated analytical platforms.

This review aims to report the state-of-the-art characteristics on the development of aptamer-based biosensors (aptasensors) [51] for the detection and quantification of allergens in food matrices using luminescent detection. The major recent achievements in this field will be reported, showing their advantages and weaknesses, as well as the probable future developments, with a critical comparison of the already available commercialized systems. 

## 2. Food Matrices and Sample Pretreatment for Portable Biosensors

Eight major food groups (milk, eggs, fish, crustacean shellfish, tree nuts, peanuts, wheat, and soybeans) are responsible for 90% of food allergies [52] and were therefore called the “eight major allergens” by the Food and Agriculture Organization of the United Nations (FAO) in 1995 [7]. However, in addition to the compounds that are naturally present in foods, it has been recently observed that allergic reactions can also be ascribed to exogenous substances, for example, to antibiotic residues that are mostly contained in milk and meat.

Food categories are vastly different from each other, both due to the nature of the allergens (usually proteins) and the characteristics of the matrix. Indeed, one of the first issues to be considered in developing an analytical protocol for food allergens is related to the sampling procedure, which must consider the physical form of the food matrix. Several foods (e.g., beverages, flours, and frozen desserts) are relatively homogeneous, and allergens are, on principle, uniformly distributed. On the contrary, other foods can show a nonhomogeneous distribution of allergens (e.g., due a particulate nature), thus posing a grave concern in obtaining a representative sample of the entire food batch [53]. Furthermore, the analysis of food products with conventional analytical techniques requires complex preanalytical treatments, often including purification and preconcentration steps. These treatments require dedicated laboratories and trained personnel for their execution, thus not being suitable for on-site analyses. On the contrary, analytical methods employing biospecific probes able to selectively recognize the target analyte in complex matrices may be coupled with extraction methods with lower performances but characterized by high simplicity and a rapidity of execution (Figure 2). Such extraction procedures will be also sustainable and environmentally friendly, e.g., will use aqueous extraction buffers, since biosensing methods are not compatible with most organic solvents [54]. Indeed, commercial LFIA kits already employ fast and simple procedures for the recovery of target analytes from different food matrices. In such procedures, solid food samples are finely homogenized and then suspended in an aqueous extraction buffer. The mixture is stirred for a few minutes, and the supernatant is collected and analyzed. The procedure for liquid food matrices is similar, except that it is sufficient to solubilize an aliquot of a sample in the extraction buffer.

The extraction procedure should recover the target analyte (allergen or food marker) while minimizing the extraction of other matrix components that may interfere with the assay. It should also be applicable both to the food as-is and to processed food derivatives, in which technological treatments may modify the characteristics of the target analyte (for example, the protein solubility can decrease because of changes in the pH, chemical modifications, or aggregation phenomena due to heat treatment, hydrolysis, fermentation, etc.). Due to the structural diversity of the targets, both native and modified, the extraction procedure (e.g., buffer components, pH, ionic strength, and temperature) must be carefully optimized. Indeed, in the specific case of the pretreatment of the sample to be analyzed with an aptamer-based analytical method, the specific molecular conformation of the target analyte in the sample, as well as the presence of interferents that can cause alterations in the aptamer folding or in the structure of the target ligand, can induce unexpected deviations in the binding affinity, affecting the overall performances of the assay [55]. Uncontrolled nonspecific interactions could influence the overall specificity and reproducibility of the methods, thus making the sample pretreatment a critical step for the analytical performances of these methods. Depending on the extraction procedure applied to food samples, the specific interaction between an aptamer and its target can be hindered by the presence of chemical or biological interferences that result in a reduction in the selectivity and sensitivity [56].

Numerous studies have shown that different extraction buffers can affect not only the quantity but also the quality of the extracted proteins, especially in the case of foods subjected to technological treatments [57,58]. Moreover, due to the complexity of food matrices, substances present in the food can negatively affect the extraction of target proteins [59]. An incorrect extraction procedure can lead both to false positives (extraction of matrix components capable of causing interference in the immunochemical assay) and false negatives (often due to insufficient extraction of the target proteins, especially in the case of processed foods), the latter being the situation that occurs more frequently. Buffers containing sodium dodecyl sulphate (SDS) and 2-mercaptoethanol (2-ME) as reducing agents have been proposed for the extraction of proteins modified by heat treatments and high pressures for several types of allergens (milk, egg, wheat, buckwheat, peanut, soy, and shellfish) [60]. Recently, Ito et al. proposed sodium sulfite as a potential eco-friendly reductant alternative to 2-ME [61]. Application of the modified extraction buffer on different matrices (egg, milk, wheat, peanut, and buckwheat) showed that the protein extraction efficiency of SDS/0.1 mol L^−1^ sodium sulfite solution was comparable to that of SDS/2-ME [62]. Dedicated analytical devices for the detection of food allergens have also been developed. Stidham et al. recently described a portable integrated system for aptamer-based allergen assays employing a single-use test pod for carrying out both a sample preanalytical treatment (i.e., homogenization and extraction) and fluorescence polarization-based analysis. This system has been applied to a wide variety of food matrices, providing a robust and sensitive detection of peanut proteins at concentrations as low as 12.5 ppm [63].

## 3. Luminescence-Based Aptasensors for Food Allergens

The development of luminescence-based biosensors represents a very prolific research area, taking advantage of the considerable recent advancements in the technologies available for luminescent signals detection. Indeed, in the integration of all the elements required for analysis within a standalone, portable, and easy-to-use analytical platform, maintaining a high sensitivity is fundamental for the development of a portable biosensor. 

Several luminescence phenomena have been exploited in miniaturized analytical devices, each of them with its own unique advantages related to the light signal-triggering process. Among these, FL and CL are the most successful in the field of biosensors and have recently been widely applied in the development of aptasensors. The remarkable success of FL-based biosensors is related to their high potential sensitivity, to the availability of many efficient FL dyes that can be used as labels, to the simple analytical procedures, and to the previous development of FL detection systems for other analytical formats. Nonetheless, these biosensors face several limitations in terms of instrumentation. Indeed, an optical module is required, comprising a light source, optics (e.g., lenses or dichroic mirrors), and wavelength selectors (e.g., interference filters). In addition, the measurement cell must meet specific requirements in terms of geometry to achieve the optimal sample excitation and emitted light collection. On the other hand, CL detection, which relies on chemical reactions able to produce photons, provides a high sensitivity due to the weakness of the background signal (CL reactions are very selective; thus, in the absence of the target analyte, no CL signal is observed). Moreover, CL measurements can be performed with very simple instrumentation (i.e., no light source or wavelength selectors are needed) and using a variety of light detectors, such as photomultipliers, a charge-coupled device (CCD) and complementary metal–oxide semiconductor (CMOS) imaging sensors, or thin-film photosensors. Furthermore, there are less requirements for sample geometry [64]. However, the addition of the chemical reagents that trigger the CL reaction requires a biosensing device equipped with a dedicated fluidic system and complicates the analytical protocol. Such drawbacks are partially removed in ECL detection, in which CL emission is triggered by the application of a suitable potential to electrodes embedded in the measurement cell rather than by the addition of a chemical reagent, providing an easier control of light emission in both space and time.

### 3.1. Fluorescent Aptasensors

Fluorescence detection technology has been widely employed for the development of aptasensors for food allergens [65]. In most cases, the aptamers were labeled with fluorescent dyes or, more recently, nanosized labels, such as quantum dots (QDs), even though assay formats employing unlabeled aptamers are also possible [66,67].

Zhang et al. developed a label-free aptamer-based strategy for detecting the major shrimp allergen tropomyosin by using the commercially available OliGreen fluorescent dye and magnetic nanoparticles (MNPs) as separation carriers (Figure 3a) [68]. The assay mechanism involved MNPs coated with capture probes hybridized with a tropomyosin-binding aptamer. When the target analyte and the aptamer interact, a conformational change occurs, resulting in the release of the aptamer from the MNPs. Then, the binding of the OliGreen dye to the tropomyosin-bound aptamer caused an enhancement of the fluorescent signal; otherwise, the aptamer remained hybridized with the capture probe, and the OliGreen dye remained in the solution, producing a much weaker fluorescence emission. Thus, the increase of the fluorescence intensity is proportional to the concentration of the target analyte. Under optimal conditions, the linear range extended from 0.4 to 5 µg mL^−1^ of tropomyosin, with a limit of detection (LOD) of 0.077 µg mL^−1^.

Based on the same approach, the authors also developed another fluorescent assay for tropomyosin in which graphene oxide (GO) was used to remove unbounded aptamers [47]. Indeed, GO is suitable for the surface adsorption of many molecules (small organic molecules, peptides, nucleic acids, and proteins), as well as bacteria, and it shows good fluorescence quenching ability with the fluorescence resonance energy transfer (FRET) mechanism [69]. The authors proposed a label-free approach in which the unfolded aptamers were adsorbed by GO, while the aptamer–analyte complexes were detected quantitatively in the solution by adding the OliGreen fluorescent reagent. The analytical performance of the assay was similar to that of the previous one: the LOD was 0.15 μg mL^−1^ of tropomyosin, and the working concentration range was from 0.5 to 50 µg mL^−1^.

The use of GO in the development of fluorescent aptasensors is widely employed, as also attested by Chinnappan et al., who described a tropomyosin biosensor in which GO was used as a platform for screening the minimal length aptamer sequence required for high-affinity target binding [70]. In this biosensor, GO adsorbed a fluorescein-labeled aptamer and quenched its emission by π-stacking interactions. After the addition of the target analyte, the fluorescence was restored due to the competitive binding of the aptamer by tropomyosin, resulting in its release from GO. The biosensor showed a LOD of 2.5 nmol L^−1^ tropomyosin, and the assay performance was assessed by analyzing tropomyosin-spiked chicken soup, proving a nearly quantitative recovery (ca. 97 ± 10%). 

Weng et al combined the properties of GO with those of quantum dots (QDs) within a one-step “turn-on” homogeneous assay for the detection of Ara h 1, one of the major peanut allergens [37]. They designed a microfluidic system in which QD–aptamer–GO complexes used as probes undergo a conformational change upon interaction with the target analyte. The interaction leads to desorption of the QD–aptamer and recovery of its fluorescence properties. Using this biosensor, they obtained a LOD of 56 ng mL^−1^ of Ara h 1. Another approach based on the use of fluorescent dots as labels was reported by Shi et al. [71]. Aptamers for β-lactoglobulin were immobilized on MNPs and complexed to a complementary sequence of DNA labeled with fluorescent carbon dots (C-dot-cDNA). In the presence of β-lactoglobulin, the competition between the target analyte and the complementary sequence of DNA for binding to the aptamer resulted in the partial release of C-dot-cDNA into the solution. After magnetic separation to remove the bound C-dot-cDNA, the fluorescence of the solution (which is proportional to the concentration of β-lactoglobulin) was measured. The assay allowed the quantification of β-lactoglobulin in the 0.25–50 ng mL^−1^ range, with a LOD of 37 pg mL^−1^. Carbon dots were also employed by Zhou et al. in an aptamer-based “on-off-on” fluorescent biosensor for the detection of the shellfish allergen arginine kinase [38]. The assay used carbon dot-labeled aptamers adsorbed on GO, which efficiently quenched their fluorescence emission. Upon the addition of arginine kinase, the aptamers were released from GO, forming the aptamer–arginine kinase complex in the solution, thus restoring the fluorescence emission (Figure 3b). This approach allowed to obtain a linear range from 1 to 10,000 ng mL^−1^ with a LOD of 0.14 ng mL^−1^ of arginine kinase. Sapkota et al. exploited the FRET mechanism in a fluorescent aptasensor for the detection of lysozymes [72]. The aptasensor consisted of two partially complementary DNA arms, each labeled with a donor or acceptor fluorophore to allow FRET when the complementary arms hybridize to each other. In the normal form (open state), the hybridization of the arms is hindered by the binding of a couple of oligonucleotide sequences, comprising a lysozyme-specific aptamer. The interaction between the lysozyme and the aptamer causes a sequence of hybridization reactions, resulting in the switching of the aptasensor to the fully hybridized form (closed state), exhibiting an efficient FRET process. The aptasensor demonstrated a high sensitivity, allowing to achieve a LOD of 30 nmol L^−1^ of lysozyme and a dynamic range extended up to ~2 µmol L^−1^, and selectivity, showing no interferences from similar biomolecules. Mairal et al. developed a dimeric aptamer FRET probe in which the oligonucleotide sequences were functionalized with both donor and acceptor dyes for the detection of β-conglutin, a lupin allergen [73]. The interaction with the target analyte caused a conformation change, enhancing the FRET process. The assay was sensitive (the LOD was about 150 pmol L^−1^ of β-conglutin), and more interestingly, it showed a remarkable rapidity, since the maximum response was reached after just 1 min of incubation. Phadke et al. used fluorogenic peptide aptamers for detecting α_s_-casein [74]. The selected aptamers instantaneously enhanced their fluorescence emission upon binding to the target molecule (the assay response time was less than one minute), and further chemical modification at the N-terminus with polyethylene glycol eliminated the interference due to β-lactoglobulin. The LOD of the fluorogenic aptamer system was about 0.04 μmol L^−1^ (ca. 1 ppm), which is comparable with that of commercially available LFIAs for α_s_-casein. Wang et al. developed a dual mode aptasensor for the detection of parvalbumin with both colorimetric and fluorescent signal readouts [46]. The aptasensor was obtained by hybridizing a gold nanoparticle-modified aptamer (AuNP-APT) with complementary short-chain oligonucleotides modified either with gold nanoparticles (AuNP-CS1) or a fluorescent dye (FAM-CS2). The interaction between APT and parvalbumin led to the disassembly of the aptasensor, which resulted in a color change or in the recovery of the previously quenched fluorescence emission for the aptasensors employing the short-chain oligonucleotides AuNP-CS1 or FAM-CS2, respectively (Figure 3c). The aptasensor allowed quantitative measurements of parvalbumin in the concentration ranges 2.5–20 and 2.38–40 μg mL^−1^ for the colorimetric and fluorescence readouts, respectively, while the LOD of the fluorescence-based assay was 0.72 μg mL^−1^. Finally, Leung et al. used a square–planar fluorescent platinum(II) complex to develop a “switch-on” assay for kanamycin (Figure 3d) [75]. This platinum(II) complex, weakly emitting in the solution, showed a strong increase of fluorescence emission upon binding to the duplex DNA via intercalation, which has been utilized as a signal transduction mechanism. The assay is based on the fact that, upon interaction with kanamycin, the aptamer undergoes a conformation change from a random-coiled structure into a new conformation containing a hairpin region. This allowed intercalation of the platinum(II) complex into the bound aptamer, thus resulting in an enhancement of the fluorescence emission. The LOD of the assay was about 140 nmol L^−1^ of kanamycin.

### 3.2. Chemiluminescent Aptasensors

To develop CL-based aptasensors, aptamers can be labeled with markers commonly used for other biospecific probes (e.g., antibodies), such as enzymes detectable with CL substrates [76,77] or metal nanoparticles able to catalyze CL reactions [78,79]. In addition, it is also possible to exploit the catalytic ability of specific nucleic acid sequences, such as the hemin/G-quadruplex horseradish peroxidase (HRP)-mimicking DNAzyme [80,81], which catalyzes the oxidation of luminol by H_2_O_2_ to yield CL emissions [82]. DNAzymes can be easily incorporated into oligonucleotide aptamers by appropriately designing their sequences [83,84,85], thus enabling the development of innovative CL aptasensors [86,87,88,89,90,91].

Despite the potential advantages of CL detection, a few CL-based aptasensors for allergenic compounds in foods have been reported to date. Furthermore, these examples mainly concern antibiotic residues, which can also cause allergic reactions, rather than food-specific allergens. Hao et al. described a CL-based aptasensor for chloramphenicol using flower-like gold nanostructures (AuNFs) for aptamer labeling [92]. The assay employed a capture probe, obtained by immobilizing a biotinylated chloramphenicol-specific aptamer on avidin-modified MNPs, and a detection probe consisting of a thiolated complementary oligonucleotide sequence conjugated to N-(4-aminobutyl)-N-ethylisoluminol (ABEI)-functionalized AuNFs. The CL signal was generated by the oxidation of ABEI by H_2_O_2_ in the presence of p-iodophenol, which acted as a CL emission enhancer (the isoluminol derivative ABEI was preferred to luminol, since it maintains a relatively high CL quantum efficiency when chemically conjugated with specific analytes). During the assay, the analyte and the detection probe compete for binding to the capture probe, followed by magnetic separation of the capture probe and the addition of H_2_O_2_ and p-iodophenol to trigger the CL reaction. The achieved LODs were 0.01 ng mL^−1^ and 1 ng mL^−1^ of chloramphenicol in buffer and milk, respectively. The same authors used a similar format to develop a multiplexed CL method for the detection of three antibiotics (oxytetracycline, tetracycline, and kanamycin) in milk [93] (Figure 4a). Aptamers specific for the target analytes, acting as capture probes, were immobilized in the wells of a microtiter plate; then, the sample was added to the wells together with detection probes consisting of complementary oligonucleotide sequences modified with ABEI-functionalized AuNFs. After the competition of the analytes and the detection probes for binding to the immobilized aptamers, the bound detection probes were detected by CL by adding H_2_O_2_ and PIP. The detection limits achieved were 0.02 ng mL^−1^ (oxytetracycline), 0.02 ng mL^−1^ (tetracycline), and 0.002 ng mL^−1^ (kanamycin). Yang et al. described a CL-based aptasensor for the detection of sulfamethazine, another common antibiotic residue in milk [94]. The assay was performed on a streptavidinated microtiter plate coated with a biotin-functionalized capture aptamer and was based on the competition between the analyte in the sample and a tracer (a sulfamethazine analog conjugated to the CL enzyme HRP) for binding to the capture aptamer, followed by CL detection of the bound tracer. Milk samples could be analyzed after a simple preanalytical treatment (i.e., the centrifugation and dilution of the supernatant with a buffer to reduce the matrix effect) achieving a LOD of 0.92 ng mL^−1^, thus making this aptasensor a promising alternative to immunosensors for the detection of sulfamethazine in food samples.

Catalytic metallic nanoparticles have also been used as CL labels, as reported by Yao et al. [95]. They designed an assay platform employing magnetic beads (MBs) and exploiting the catalytic properties of gold nanocluster (AuNC) towards the luminol-H_2_O_2_ CL reaction. Aptamers specific for the target analyte (kanamycin) were immobilized on MBs and hybridized with a complementary oligonucleotide sequence labeled with AuNC. In the presence of the target analyte, its interaction with the aptamer resulted in the release of the AuNC-labeled oligonucleotide sequences. The MBs were removed by magnetic separation; then, the released oligonucleotide sequences were detected in the solution thanks to the enhancement of the CL emission due the AuNC labels, achieving a LOD of 0.035 nmol L^−1^ of kanamycin. Recently, Yan et al. proposed a label-free dual aptasensor for the detection of ATP and chloramphenicol in food matrices [96]. Oligonucleotide capture probes for ATP- and chloramphenicol-binding aptamers were immobilized on polystyrene and magnetic microspheres, respectively. Then, the microspheres were incubated with the sample in the presence of the analyte-binding aptamers. The competition between the analytes and the immobilized capture probes for binding to the aptamers resulted in amounts of aptamer bound to the microspheres that are inversely proportional to the analyte concentrations. The microspheres were separately collected by magnetic separation and centrifugation; then, the bound aptamers were detected thanks to the CL reaction of the guanine DNA nucleobase with phenylglyoxal and N,N-dimethylformamide (Figure 4b). The dual aptasensor exhibited high selectivity and sensitivity, with LODs of 37.6 and 24.8 nmol L^−1^ for ATP and chloramphenicol, respectively. Furthermore, the authors pointed out the potential application of the guanine CL reaction as a new detection strategy in unlabeled CL assays.

It should be pointed out that several colorimetric aptasensors for food allergens described in the literature employ labels (e.g., HRP) that are also suitable for CL detection [97]. Such assays could be easily adapted to this detection technology, taking advantage of its superior characteristics, e.g., the higher detectability of the CL signal. 

### 3.3. Electrochemiluminescent Aptasensors

As for CL aptasensors, only a few aptamer-based assays for food allergens with ECL detection have been reported in the literature. Du et al. developed a lysozyme aptasensor using Ru(bpy)_3_^2+^-silica@poly-L-lysine-Au (RuSiNPs@PLL-Au) nanocomposites as labels [98]. A 3D graphene-modified electrode was coated with AuNPs, then functionalized with a lysozyme-binding aptamer hybridized with a complementary single-strand DNA sequence labeled by RuSiNPs@PLL-Au, which acted as an ECL signal amplifier. In the presence of a lysozyme, the complementary DNA sequence of the duplex was displaced by the lysozyme, resulting in weaker ECL emissions. The decrease in ECL emission intensity was proportional to the logarithmic concentration of the lysozyme in the concentration range 2.25 × 10^−12^ to 5.0 × 10^−8^ mol L^−1^, and the LOD of the assay was estimated as 7.5 × 10^−13^ mol L^−1^. The aptasensor was tested in wine samples, obtaining a good agreement with the reference analytical methods, making it suitable for application in food safety monitoring. Huang et al. also described an aptasensor for lysozymes employing QDs as ECL labels [42]. The sample was first incubated with lysozyme-specific capture aptamers immobilized on an Au electrode; then, the free aptamers were hybridized with biotin-modified single-stranded oligonucleotides. Finally, an avidin-QD tracer was bound to the hybridized oligonucleotides through the biotin-avidin system. The ECL signal, which was proportional to the amount of bound QDs, was then measured in the presence of the co-reactant S_2_O_8_^2-^. The assay was not applied to food matrices, but it might be of interest due to its target (in addition to ovalbumin, a lysozyme is the major allergenic protein in egg whites, and it is also frequently used as a preservative by the food industry).

## 4. Conclusions

The development of rapid, sensitive, and low-cost allergen detection methods for food analysis outside centralized laboratories and by untrained personnel is an open issue in the food industry. Nowadays, such analyses are performed primarily by means of immunoassays (i.e., LFIAs), while aptasensors, especially in combination with sensitive luminescence-based detection techniques, are still little-explored, even if they have the potential to fulfil this need (see Table 3 and Table 4). Indeed, the increasingly advanced technologies for the detection of photons would allow to develop portable, fast, easy-to-use, and low-cost biosensors, allowing their wide diffusion as the reference method.

In addition, aptamers could allow to overcome some limitations of antibodies, such as their sensitivity to temperature and pH changes, the long and complex production procedures, and the limited possibility of chemical modification, e.g., to enable easy immobilization or labeling.

Most of the aptasensors reported here have been developed and tested only in laboratory settings, and the number of assays exploiting CL or ECL detection is quite low in comparison to those relying on FL measurements. Nevertheless, CL and ECL aptasensors have already proved to be powerful detection techniques, e.g., in clinical chemistry analyses, and are more amenable to implementation in miniaturized analytical devices than FL aptasensors. Furthermore, as stated earlier, colorimetric aptasensors employing labels such as HRP, HRP-mimicking DNAzymes, or catalytic metal nanoparticles could be easily adapted to CL or even ECL detection. Other luminescence-based detection principles could also be investigated. For example, thermochemiluminescence (TCL) detection, which employs as labels thermally unstable molecules that decompose with light emission upon heating, has been recently proposed as a suitable detection method for bioassays [99]. A few examples of TCL-based portable bioassays have been reported [100], and their application in the detection of foodborne allergens could provide significant advantages. Indeed, this detection principle combines the advantages of CL (i.e., a high detectability due to the absence of background signal), FL (i.e., no need to add further chemicals for detection), and ECL (i.e., the emission could be easily switched on or off by controlling the temperature). 

In conclusion, we expect that the continuing advances in the fields of luminescent substrates and systems, nanomaterials, and sensitive detectors will lead to an ever-greater diffusion of CL/ECL aptasensors. A further necessary step will be an advancement in the design and implementation of aptasensors in integrated, portable analytical platforms, as well as the development of simple sample treatment procedures. With the continuous efforts of scientific researchers, this will lead to easy-to-use bioassays suitable for the evaluation of the unintentional contamination of foodstuffs with allergens in the various stages of the production chain that can be used in routine analyses.

## Figures and Tables

**Figure 1 biosensors-12-00644-f001:**
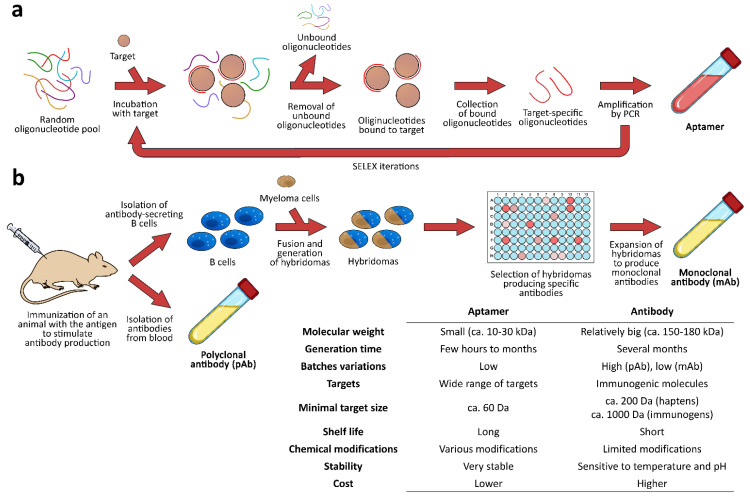
Processes for producing (**a**) aptamers and (**b**) antibodies and a comparison of their main characteristics.

**Figure 2 biosensors-12-00644-f002:**
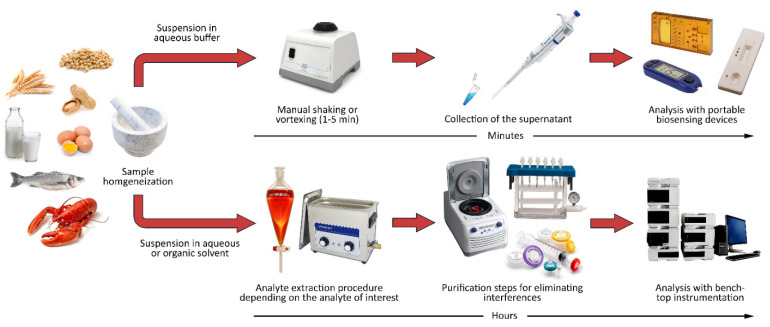
Preanalytical procedures needed for bioassays and for conventional laboratory analyses.

**Figure 3 biosensors-12-00644-f003:**
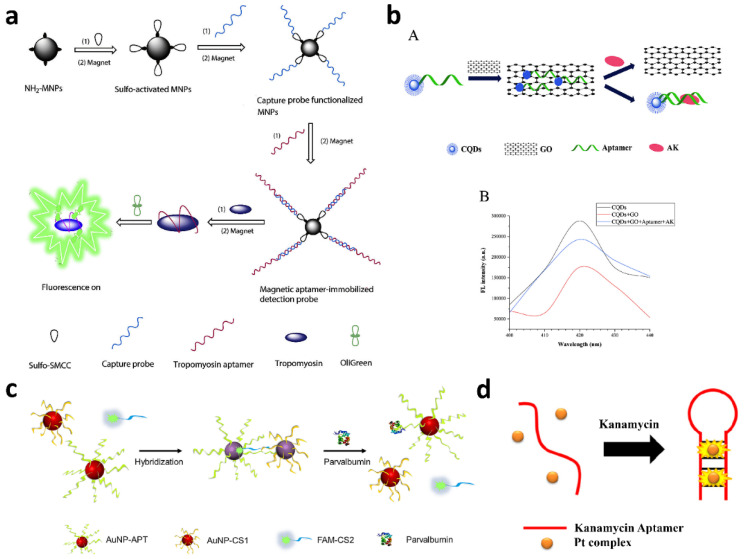
(**a**) Schematic illustration of the magnetic-assisted fluorescent aptamer assay for tropomyosin detection. Reprinted with permission from Ref. [68]). Copyright 2022, Elsevier. (**b**) (**A**) Schematic illustration of the “on-off-on” fluorescence aptasensor for adenine kinase detection; (**B**) Recovery of fluorescence intensity upon the addition of shellfish AK and the consequent release of the cCQD-aptamer from the GO surface forming cCQD-aptamers-AK complex. Reprinted with permission from Ref. [38]). Copyright 2022, Elsevier.(**c**) Schematic illustration of the dual-mode aptasensor for parvalbumin . Reprinted with permission from Ref. [46])]. Copyright 2022, Elsevier. (**d**) Schematic illustration of the platinum(II) complex-based “switch-on” fluorescent assay for kanamycin. Reprinted with permission from Ref. [75]. Copyright 2022, Elsevier.

**Figure 4 biosensors-12-00644-f004:**
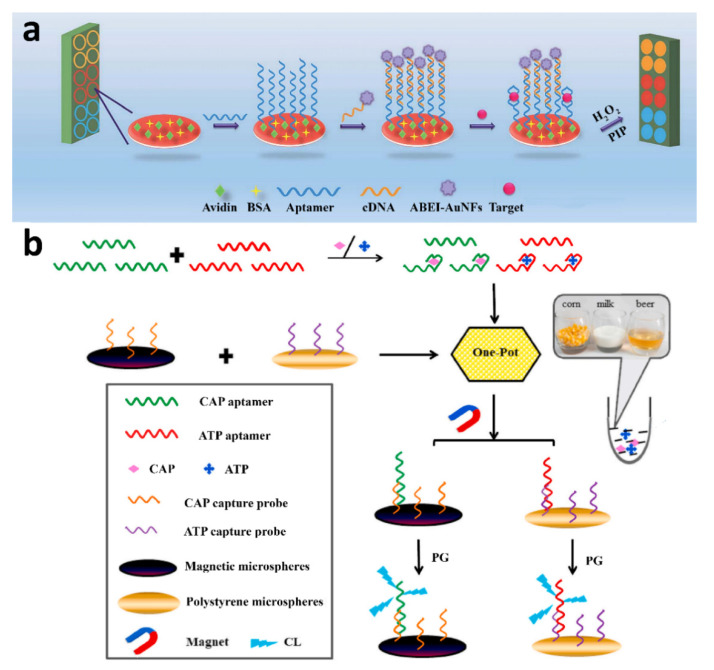
(**a**) Schematic illustration of the multiplex antibiotic detection based on the aptamer-modified ABEI-AuNFsReprinted with permission from Ref. [93]). Copyright 2022, Royal Society of Chemistry. (**b**) Schematic representation of the label-free CL aptasensor for the simultaneous detection of ATP (adenosine triphosphate) and chloramphenicol. Reprinted with permission from Ref. [96]. Copyright 2022, Elsevier.

**Table 1 biosensors-12-00644-t001:** Commercially available LFIAs for food allergens.

Commercial Name	Allergen or Food	Limit of Detection (ppm)	Company	Reference
3M Rapid Kit	Almond	2	3M (Saint Paul, MN, USA)	[20]
	Cashew	2		
	Coconut	2		
	Egg	0.5		
	Fish	1		
	Gluten	5		
	Hazelnut	2		
	Milk	3		
	Peanut	1		
	Pecan	3		
	Pistachio	2		
	Soy	2		
	Walnut	2		
Agitest	Almond	1	Rega Biotechnology Inc. (New Taipei City, Taiwan)	[21]
	Buckwheat	1		
	Casein	100		
	Egg	1		
	Fish	0.1		
	Gluten	20		
	Mango	2		
	Peanut	1		
	Sesame	0.2		
	Shellfish	1		
	Soy	10		
AgraStrip	Almond	2	Romer Labs GmbH (Getzersdorf, Austria)	[22]
	B-Lactoglobulin	0.5		
	Brazil Nut	5		
	Casein	1		
	Cashew/Pistachio	2		
	Crustacean	2		
	Coconut	10		
	Gluten	4		
	Hazelnut	5		
	Lupin	10		
	Macadamia Nut	2		
	Milk	1		
	Mustard	2		
	Peanut	1		
	Sesame	5		
	Soy	2		
	Walnut	10		
	Whole Egg			
AlerTox Sticks	Almond	20	Hygiena LLC (Camarillo, CA, USA)	[23]
	Β-Lactoglobulin	2.5		
	Casein	2.5		
	Crustacean	10		
	Egg	1.25		
	Fish	5		
	Hazelnut	20		
	Milk	2.5		
	Mustard	2		
	Peanut	1		
	Sesame	3		
	Soy	10		
	Walnut	2.25		
Aller-ROSA	Milk	2–5	Charm Sciences Inc. (Lawrence, MA, USA)	[24]
Reveal/Reveal 3D	Almond	1	Neogen Co. (Lansing, MI, USA)	[25]
	Coconut	1		
	Crustacean	1–5		
	Egg	2.4		
	Gliadin	5		
	Gluten	5–10		
	Hazelnut	0.75–1.5		
	Milk	2		
	Multi-Tree nuts	1–2		
	Mustard	1.3		
	Peanut	1.3		
	Sesame	1		
	Soy	2.5		
SENSIStrip	Almond	1	Eurofin Technologies (Budapest, Hungary)	[26]
	Casein	20		
	Shellfish	1		
	Egg	1		
	Fish	1		
	Peanut	1		
	Soy	10		
	Gluten	2		

**Table 2 biosensors-12-00644-t002:** Selection of oligonucleotide aptamers for food allergens reported in the literature.

Allergen	Aptamer Sequence	Reference
Ara h 1 ^1^	(5’) TCG CAC ATT CCG CTT CTA CCG GGG GGG TCG AGC GAG TGA GCG AAT CTG TGG GTG GGC CGT AAG TCC GTG TGT GCG AA (3’)	[37]
Arginine kinase ^2^	(5’) GGC GAA CAG CAG CGC GAT TCG GGT TGC GGA TAG TGA CAT A (3’)	[38]
β-Conglutin ^3^	(5’) AGC TGA CAC AGC AGG TTG GTG GGG GCT TCC AGT TGG GTT GAC AAT ACG TAG GGA CAC GAA GTC CAA CCA CGA GTC GAG CAA TCT CGA AAT (3’)	[39]
Gluten ^4^	(5’) CCA GTC TCC CGT TTA CCG CGC CTA CAC ATG TCT GAA TGC C (3’)	[40]
	(5’) CTA GGC GAA ATA TAG CTA CAA CTG TCT GAA GGC ACC CAA T (3’)	[40]
β-Lactoglobulin ^5^	(5’) CGA CGA TCG GAC CGC AGT ACC CAC CCA CCA GCC CCA ACA TCA TGC CCA TCC GTG TGT G (3’)	[41]
Lysozyme ^6^	(5’) ATC TAC GAA TTC ATC AGG GCT AAA GAG TGC AGA GTT ACT TAG (3’)	[42]
	(5’) ATC AGG GCT AAA GAG TGC AGA GTT ACT TAG (3’)	[43]
	(5’) GGG AAT GGA TCC ACA TCT ACG AAT TCA TCA GGG CTA AAG AGT GCA GAG TTA CTT AGT TCA CTG CAG ACT TGA CGA AGC TT (3’)	[44]
	(5’) GCA GCT AAG CAG GCG GCT CAC AAA ACC ATT CGC ATG CGG C (3’)	[45]
Parvalbumin ^7^	(5’) GCC AAA GGA GGC GAG AGA TAA AAG ATT GCG AAT CCA TTC G (3’)	[46]
Tropomyosin ^8^	(5’) TAC TAA CGG TAC AAG CTA CCA GGC CGC CAA CGT TGA CCT AGA AGC ACT GCC AGA CCC GAA CGT TGA CCT AGA AGC (3’)	[47]

^1^ Peanut. ^2^ Crustaceans. ^3^ Lupin. ^4^ Wheat, barley, and rye. ^5^ Milk. ^6^ Egg whites. ^7^ Fish. ^8^ Crustaceans and mollusks.

**Table 3 biosensors-12-00644-t003:** Scheme of the luminescent aptasensors for food allergen detection reported in the literature.

Detection Method	Mechanism	Label	Analyte	Detection Limit	Ref.
**Fluorescence** Advantages: ✓ high potential sensitivity ✓ availability of many efficient FL dyes that can be used as labels ✓ simple analytical procedures ✓ previous development of FL detection systems for other analytical formats Disadvantages: - an optical module is required, comprising a light source, optics (e.g., lenses or dichroic mirrors), and wavelength selectors (e.g., interference filters) - the measurement cell must meet specific requirements in terms of geometry to achieve optimal sample excitation and emitted light collection	The capture aptamer was conjugated on the surface of MNPs. When the aptamer interacts with target analytes, it was released from the surface of MNPs, thus producing a fluorescent signal by adding the OliGreen dye, which is able to enhance its fluorescence upon binding to ssDNA.	Label-free	Tropomyosin	0.077 µg mL^−1^	[68]
	Label-free fluorescent approach was exploited by utilizing the OliGreen ssDNA reagent to quantitatively detect the aptamers bound to analyte in solution with the aid of the adsorption of unfolded aptamers by GO.	Label-free	Tropomyosin	0.15 μg mL^−1^	[69]
	A fluorescein dye-labeled GO quenches the truncated DNA aptamer. After the addition of the target analyte, the fluorescence was restored due to the competitive binding of the aptamer to GO.	Fluorescein dye	Tropomyosin	2.5 nmol L^−1^	[70]
	The formation of QD-DNA aptamer–GO complexes as probes is able to undergo conformational change upon interaction with the target analytes, resulting in fluorescence changes: fluorescence is quenched or recovered depending on the adsorption and desorption of aptamer-QDs on GO.	QDs	Ara h 1	56 ng mL^−1^	[37]
	The aptamer was immoblized on MNPs, and the C-dots served as a label for the cDNA. The aptamer preferentially binds the target analyte, leading to a partial release of the C-dots-cDNA into the solution. After magnetic separation, the solution contained the released C-dots-cDNA, which are quantified by fluorescence.	C-dots	β-lactoglobulin	37 pg mL^−1^	[71]
	QDs-DNA aptamer probe and GO were self-assembled to effectively quench the fluorescence of the Qdots. Upon adding the target analyte, the QDs-aptamer was released from the GO surface and formed the QDs–aptamers–analyte complex, leading to a fluorescent signal.	QDs	Arginine kinase	0.14 ng mL^−1^	[38]
	The aptamer is composed of two partially cDNA arms, each labeled with either a donor (Cy3) or an acceptor (Cy5) fluorophore to enable FRET when the complementary arms hybridize to one another.	Donor (Cy3) and acceptor (Cy5) fluorophore to enable FRET	Lysozyme	30 nmol L^−1^	[72]
	The probe was represented by a dimeric aptamer, with each monomeric aptamer being flanked by donor/acceptor moieties. Upon addition of target analyte, the specific interaction induces a change in the biaptameric structure, resulting in an increase in fluorescence emission.	Donor (Alexa Fluor 488) and acceptor (Alexa Fluor 555) fluorophore to enable FRET	β-conglutin	150 pmol L^−1^	[73]
	The assay was based on the use of two fluorogenic peptide aptamers that instantaneously enhance their fluorescence upon binding to a target molecule.	Label free	α_s_-casein	0.04 μmol L^−1^	[74]
	The aptasensor was based on hybridization of the DNA aptamer-modified AuNP, the complementary short chain-modified gold nanoparticles and the fluorescent dye-labeled complementary short chain. The presence of target analyte led to a competition, which allows to observe a change in the solution color of the AuNPs and a recovery of the fluorescence signals of FAM-CS2.	AuNP (colorimetric detection) and Fluorescent dye (fluorescent detection)	Parvalbumin	0.72 μg mL^−1^	[46]
	The assay was based on a square–planar luminescent platinum(II) complex and the DNA aptamer. Upon the addition of the target analyte, the aptamer changes from a random-coiled structure into a specific conformation containing a hairpin region, allowing the intercalation of the platinum(II) complex into the bound aptamer and enhancing the luminescence signal.	Label-free	Kanamycin	140 nmol L^−1^	[75]
**Chemiluminescence** Advantages: ✓ high sensitivity due to the weakness of the background signal ✓ CL measurements can be performed with very simple instrumentation (i.e., no light source or wavelength selectors are needed) and using a variety of light detectors ✓ No requirements for sample geometry Disadvantages: - The addition of the chemical reagents that trigger the CL reaction requires a biosensing device equipped with a dedicated fluidic system and complicates the analytical protocol.	The assay employed a capture probe, obtained by immobilizing a biotinylated chloramphenicol-specific aptamer on avidin-modified MNPs, and a detection probe consisting of c-DNA sequence-conjugated ABEI-functionalized AuNFs. The analyte and the detection probe compete for binding to the capture probe, followed by magnetic separation of the capture probe and the addition of CL substrate to trigger the CL reaction.	ABEI-AuNFs	Chloramphenicol	0.01 ng mL^−1^	[92]
	Aptamers specific for the target analytes, acting as capture probes, were immobilized in the wells of a microtiter plate; then, the sample was added to the wells together with detection probes consisting of c-DNA modified with ABEI-functionalized AuNFs. After the competition of the analytes and the detection probes for binding to the immobilized aptamers, the bound detection probes were detected by CL.	ABEI-AuNFs	Oxytetracyclin, tetracycline and kanamycin	0.02 ng mL^−1^ (oxytetracycline), 0.02 ng mL^−1^ (tetracycline) and 0.002 ng mL^−1^ (kanamycin).	[93]
	The assay was performed in a streptavidinated microtiter plate coated with a biotin-functionalized capture DNA aptamer and was based on the competition between the analyte in the sample and a tracer (a sulfamethazine ana-log conjugated to the CL enzyme HRP) for binding to the capture aptamer, followed by CL detection of the bound tracer.	HRP	Sulfamethazine	0.92 ng mL^−1^	[94]
	DNA aptamers specific for the target analyte were immobilized on MBs and hybridized with a complementary oligonucleotide sequence labeled with AuNC. In the presence of the target analyte, its interaction with the aptamer resulted in the release of the AuNC-labeled oligonucleotide sequences. The MBs were removed by magnetic separation; then, the released oligonucleotide sequences were detected.	AuNCs	Kanamycin	0.035 nmol L^−1^	[95]
	Oligonucleotide capture probes for ATP- and chloramphenicol-binding aptamers were immobilized on polystyrene and magnetic microspheres, respectively. The competition between the analytes and the immobilized capture probes for binding to the aptamers resulted in amounts of aptamer bound to the microspheres that are inversely proportional to the analyte concentrations. The bound aptamers were detected thanks to the CL reaction of the guanine DNA nucleobase with phenylglyoxal and N,N-dimethylformamide.	Label free	ATP and chloramphenicol	37.6 nmol L^−1^ (ATP) and 24.8 nmol L^−1^ (chloramphenicol)	[96]
**Electrochemiluminescence** Advantages: ✓ The CL emission is triggered by the application of a suitable potential to electrodes embedded in the measurement cell rather than by the addition of a chemical reagent, providing an easier control of light emission in both space and time. Disadvantages: - Instability of electrode (degradation biosensing reagents and the instable optical signal of ECL luminophores).	A 3D graphene-modified electrode was coated with AuNPs, then functionalized with a lysozyme binding aptamer hybridized with a complementary single-stranded DNA sequence labeled by RuSiNPs@PLL-Au, which acted as an ECL signal amplifier. In the presence of lysozyme, the cDNA sequence of the duplex was displaced by lysozyme, resulting in weaker ECL emission.	RuSiNPs@ PLL-Au	Lysozyme	7.5 × 10^−13^ mol L^−1^	[98]
	Sample was incubated with probes immobilized at Au electrode in order to form the aptamer–lysozyme bioaffinity complexes, and the free probes were hybridized with the biotin modified cDNA oligonucleotides to form double-stranded DNA (ds-DNA) oligonucleotides. Avidin-QDs were bound to these hybridized cDNA through the biotin–avidin system. The ECL signal of the biosensor was responsive to the amount of QDs bonded to the cDNA oligonucleotides, which was inverse proportional to the combined target protein.	QDs	Lysozyme	Not reported	[42]

**Table 4 biosensors-12-00644-t004:** Comparison between the analytical performance (in terms of LOD) of luminescence-based aptasensors and commercially available LFIAs for selected food allergens.

Food Allergen	Aptasensor		LFIA	
	LOD ^1^	Ref.	LOD ^2^	Ref.
Ara h 1	56 µg L^−1^	[37]	0.5 mg L^−1^ (detects Ara h 1, Ara h 2, and Ara h 3)	[23]
Casein	1 mg L^−1^	[74]	0.3 mg L^−1^	[21]
			0.25 mg L^−1^	[23]
			1.8 mg L^−1^	[25]
			0.03 mg L^−1^	[26]
β-Lactoglobulin	37 ng L^−1^	[71]	0.25 mg L^−1^	[23]
Tropomyosin	77 µg L^−1^	[68]	1.7 µg L^−1^	[26]
	0.15 mg L^−1^	[69]		
	90 µg L^−1^	[70]		

^1^ LOD values expressed in mole units were converted using the molecular weight of the target allergen protein. ^2^ In commercial LFIA assays, target allergens are often referred to as generic “food proteins”. To make the comparison significant, here, we reported only LFIAs in which the allergen protein is clearly specified, and a LOD for the protein in the solution is given or can be estimated from the food sample treatment protocol (it has been assumed that the extraction of the protein from the food matrix was quantitative).

## Data Availability

Not applicable.

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
