# Peer review of "Luminescent Aptamer-Based Bioassays for Sensitive Detection of Food Allergens"

_biosensors, 2022, doi:10.3390/bios12080644_

Round 1

Reviewer 1 Report

This review mainly introduces bioassays based on aptamer luminescence detection in the field of food allergen analysis in recent years, and the overall writing structure is clear. In my opinion, this article can be accepted after revision.

1. It is necessary to supplement the commercial detection methods of food allergens in Table 1 and increase the detection concentration of each detection method. 2. In the third section "3, luminescence-based food allergen aptamer sensors" should be added to the table summary, which involves detection methods, analyte types, detection limits, advantages, disadvantages, etc. 3. In section 3 "3. Luminescence-based food allergen aptamer sensor", there are few references in the past five years, it is recommended to supplement the latest relevant references. 4. In figure 3, the review map needs further processing and beautification. 5. Please refer to the bibliography format according to the requirements of the journal Biosensor.

Reviewer 2 Report

The review is well written. Many readers will understand the progress of aptamer sensor research and their application for the detection of contaminations in some foods.

Author Response

We thank the Reviewer for his positive evaluation of our manuscript.

Reviewer 3 Report

In the current manuscript, the authors present a comprehensive review on the recent progress in the development of luminescent aptamer-based analytical technologies for rapid and sensitive detection of allergens in foods. A brief introduction on the existing strategies of biosensing food allergens and the advantages of luminescent aptamers over other biorecognition reagents is presented. Then, the review takes a small detour, by discussing on sample preparation, a key step in food-related bioassays. After that, the authors describe extensively on the recent examples of luminescent aptamer-based biosensors, organized based on different luminescence phenomena (fluorescence, chemiluminescence and electrochemiluminescence) and focused on their mechanisms of action and limit of detection. Eventually, the review is concluded with future perspectives on this field.

Overall, the field of aptamer-based biosensor developments have advanced excitingly in the past few years, and there await more opportunities in tool developments for broader applications, especially in food safety related biosensing. Thus, by summarizing the latest progress, this review is a timely contribution to this rapidly developing field. I suggest accepting the work after the following minor concerns are addressed:

1.     Although “food matrices and sample pre-treatment” is an important topic for the application of biosensing food allergens, the connection between part 2 and the rest of the review is very loose. It seems off-the-topic because the considerations in sample preparation are not specific to aptamer-based biosensing methods. If the authors would like to keep this section, they should address whether it makes a difference for sample preparation when aptamers are used as the bioassay reagent.

2.     For section 3, the description of examples of luminescent aptamer-based biosensors are very detailed and knowledgeable. However, there lack a clear organization and connection between these examples. For instance, one method was introduced in the paragraph from line 226 to 237, while more than 10 other methods were covered in a single long paragraph from line 246 to 322. Why is the MNPs + OliGreen strategy so special? How do these methods compare to each other? What are the main differences between them and what are the pros and cons? This issue also applies to the paragraphs from line 332 to 365, from 372 to 394. For a good review, the authors shouldn’t just present the examples without a clear logic flow. These examples could be organized into several groups, and the similarities and differences between different groups could be summarized, which could help the readers better understand these recent progress in the field.     

3.     When a specific aptamer is referred, it should be made clear which type it is (DNA, RNA or peptide)

4.     The word “aptasensor” needs to be defined/explained. It may not be a common concept for the readers that are not familiar with the field.

Round 2

Reviewer 1 Report

The manuscript can be accepted in current version